# Antibiotic and Heavy Metal Susceptibility of Non-Cholera *Vibrio* Isolated from Marine Sponges and Sea Urchins: Could They Pose a Potential Risk to Public Health?

**DOI:** 10.3390/antibiotics10121561

**Published:** 2021-12-20

**Authors:** Wellington Felipe Costa, Marcia Giambiagi-deMarval, Marinella Silva Laport

**Affiliations:** Instituto de Microbiologia Paulo de Góes, Universidade Federal do Rio de Janeiro, Av. Carlos Chagas Filho, 373, Cidade Universitária, Rio de Janeiro 21941-902, Brazil; wellfecosta@micro.ufrj.br (W.F.C.); marciagm@micro.ufrj.br (M.G.-d.)

**Keywords:** beta-lactamase, copper resistant, *Darwinella*, hemolysin, *Paracentrotus lividus*, *Vibrio* alginolyticus, *Vibrio* harveyi

## Abstract

*Vibrio* is an important human and animal pathogen that can carry clinically relevant antibiotic resistance genes and is present in different aquatic environments. However, there is a knowledge gap between antibiotic and heavy metal resistance and virulence potential when it is part of the microbiota from marine invertebrates. Here, we aimed to evaluate these characteristics and the occurrence of mobile genetic elements. Of 25 non-cholera *Vibrio* spp. from marine sponges and sea urchins collected at the coastlines of Brazil and France analyzed in this study, 16 (64%) were non-susceptible to antibiotics, and two (8%) were multidrug-resistant. Beta-lactam resistance (*bla*_SHV_) and virulence (*vhh*) genes were detected in sponge-associated isolates. The resistance gene for copper and silver (*cusB*) was detected in one sea urchin isolate. Plasmids were found in 11 (44%) of the isolates. This new information allows a better comprehension of antibiotic resistance in aquatic environments, since those invertebrates host resistant *Vibrio* spp. Thus, *Vibrio* associated with marine animals may pose a potential risk to public health due to carrying these antibiotic-resistant genes.

## 1. Introduction

The genus *Vibrio* is a group of gram-negative bacilli possessing a curved-rod shape. They are ubiquitously distributed in aquatic environments, such as coastal seawater, sediments and estuaries, and are associated with marine invertebrate organisms [1]. Some species can be pathogenic for humans and aquatic animals (e.g., *Vibrio cholerae*, *Vibrio alginolyticus*, *Vibrio parahaemolyticus*, *Vibrio vulnificus* and *Vibrio harveyi*) [1,2]. In humans, *Vibrio* can cause mild to severe gastroenteritis, extra-gastrointestinal infections and sepsis, which can be fatal [2]. The bacteria are generally susceptible to antibiotics used in medical and veterinary practice. However, an increased number of resistant strains, non-cholera *Vibrio*, have been observed around the world [3].

Antibiotic-resistant *Vibrio* spp. strains, carrying clinically relevant antibiotic resistance genes (ARGs), have been isolated from marine environments [4,5,6]. Sometimes, these bacteria can also harbor virulence genes that increase their pathogenic potential [7,8]. These ARGs and virulence genes can be present on mobile genetic elements associated with heavy metal resistance genes (HMRGs). This mechanism allows the co-selection of bacteria carrying all three, ARGs, virulence genes and HMRGs, and facilitates their persistence and dissemination in environments contaminated with antibiotics and heavy metals [5,6,7,8,9].

*Vibrio* spp. is an essential member of the microbiota from marine sponges and sea urchins [10,11]. Information on the antibiotic resistance status of *Vibrio* spp. isolated from these invertebrates is still scarce. The “One Health” concept claims that the health of humans, animals and the environment be intrinsically connected [12]. Considering the co-selection potential of resistance and virulence mechanisms, marine invertebrates could be a source of non-cholera *Vibrio* strains that carry relevant (and maybe new) antibiotic resistance traits, which may contribute to the current global crisis in antibiotic resistance. On this background, we aimed to investigate if *Vibrio* associated with marine sponges and sea urchins could pose a potential risk to public health by characterizing the *Vibrio* spp. isolates phenotypically and genotypically in terms of antibiotic resistance, as well as the occurrence of heavy metal resistance genes, virulence genes and mobile genetic elements.

## 2. Results

### 2.1. Identification of Vibrio spp.

Of the 25 *Vibrio* spp. strains analyzed in this study, 19 were isolated from the Brazilian marine sponges *Darwinella* spp., *Haliclona* sp. and *Plakina cyanorosea* [13], and six from the gastrointestinal tract of the French sea urchin *Paracentrotus lividus* [10]. Previously, 14 strains were identified by 16S rRNA sequencing as *V. alginolyticus* and one as *V. harveyi* [10,13]. MALDI-TOF MS analysis confirmed the identification of 10 (67%) of the 15 potential pathogenic strains, with confidence at the genus level and likelihood at the species level. The remaining five (33%) strains showed scores consistent with probable genus classification. However, the other strains (*n* = 10) identified as belonging to four distinct *Vibrio* species or only to the *Vibrio* genus by 16S rRNA sequencing, using MALDI-TOF-MS, were identified as belonging only to three probable species. Among them were *Vibrio parahaemolyticus* (*n* = 2; 20%), *Vibrio harveyi* (*n* = 1; 10%) and *Vibrio mytili* (*n* = 1; 10%). Six (60%) strains were not reliably identified (Appendix A).

All species considered to be opportunistic human pathogens, such as *V. alginolyticus* and *V. harveyi*, were isolated from the marine sponges *Darwinella* spp. (sp.1 and sp. 2), *Haliclona* sp. and *P. cyanorosea*. The other species are rarely associated with human infection and were isolated from the marine sponges *Darwinella* sp. 2 and *Haliclona* sp. and from the sea urchin *Paracentrotus lividus*.

### 2.2. Antibiotic Non-Susceptibility among Vibrio spp. Isolates

A total of 16 (64%) *Vibrio* spp. isolates were non-susceptible (resistant and/or intermediate) to one or more antibiotics, with two (8%) strains of *V. alginolyticus* showing a MDR phenotype (Figure 1 and Appendix A). All non-susceptible strains were isolated from marine sponges only.

The highest resistance rates found in the *Vibrio* spp. strains were for cephalexin (*n* = 9, 36%) and erythromycin (*n* = 7, 28%). Non-susceptibility was also observed for amikacin (*n* = 3, 12%), ampicillin-sulbactam (*n* = 3, 12%) and sulfamethoxazole-trimethoprim (*n* = 2, 8%), as well as for tobramycin, gentamicin, ofloxacin, ciprofloxacin, cefoxitin and amoxicillin-clavulanic acid (*n* = 1, 4% for each one) (Figure 1).

### 2.3. Detection of Genotypes: Antibiotic and Heavy Metal Resistance Genes, Virulence Genes, and Integrons

Among the four classes of antibiotics (aminoglycosides, β-lactams, macrolides and quinolones) investigated in this study, we found only one gene out of a total of 18 different targets of importance in medical practice. The antibiotic resistance gene *bla*_SHV_, encoding a beta-lactamase, was detected in a single strain (*V. alginolyticus* 84BHI10) isolated from *Darwinella* sp. 2. With regard to the four heavy metal resistance genes tested, only *cusB*, which encodes resistance to copper and silver, was detected in just one isolate (*Vibrio oceanisediminis* P1I-2) from the *P. lividus* sea urchin. The virulence gene *vhh*, encoding a hemolysin, was detected in just one strain (*Vibrio* sp. ME7) isolated from *Haliclona* sp. (Figure 1 and Appendix A). The *intI1* gene, encoding the integrase from Class 1 integron, was not detected among the strains.

### 2.4. Plasmid Profile

Plasmids were found in 11 (44%) strains, which carried between one and eight different plasmids with sizes varying from 2.4 to 45.5 kb (Figure 1, Appendix A: Appendix A and Appendix A). These mobile genetic elements occurred more frequently (100%, *n* = 6/6) in the French sea urchin isolates than in isolates from the Brazilian marine sponges (26.3%, *n* = 5/19).

### 2.5. Association between Antibiotic Resistance Phenotypes and Occurrence of Plasmids

Correlation analysis of the presence/absence of antibiotic resistance phenotype and presence/absence of plasmids for the 25 *Vibrio* spp. isolates indicated a moderate inverse correlation (r = −0.5014; *p* < 0.05) of these characteristics. When considering only the 15 potentially pathogenic strains (*V. alginolyticus* and *V. harveyi*), no correlation was observed between the antibiotic resistance phenotype and the occurrence of plasmids (r = −0.0753; *p* > 0.05).

## 3. Discussion

In this study, we aimed to investigate the possibility that *Vibrio* spp. associated with marine sponges and sea urchins may pose a risk for public health. With this objective, the non-cholera *Vibrio* strains were reidentified by MALDI-TOF MS and characterized phenotypically and genotypically with regard to antimicrobial resistance, as well as to the occurrence of heavy metal resistance genes, virulence genes and mobile genetic elements.

Currently, there are 157 validated *Vibrio* species according to the Genome Taxonomy Database (GTDB; available online: https://gtdb.ecogenomic.org/stats/r95; accessed on 09 December 2021). From these, *Vibrio cholerae, Vibrio vulnificus, Vibrio parahaemolyticus* and *Vibrio alginolyticus* are the four species, also known as the “big four”, often associated with human infections [2]. In this study, among the six distinct species that were previously identified [10,11,12,13], two species were confirmed by MALDI-TOF MS. Identification and discrimination of closely related *Vibrio* spp. can be difficult and some strategies are limited in terms of the identification of environmental species [14,15]. MALDI-TOF-MS is primarily designed for clinical use, and thus, the Biotyper database mainly contains clinically important species [16]. The performance of this method depends on the strain catalogue in the reference library. However, the Bruker standard library satisfied the requirements of our work proposal to confirm the identification of potentially pathogenic species.

The role of the marine environment in the development and dissemination of antibiotic resistance is largely unknown. *Vibrio* spp. are indigenous to the sea, and in recent years, the occurrence of resistance genes in several species has been observed [6,15]. Herein, we observed that *V. alginolyticus*, previously described as a potential pathogen, was the most frequent species with an antibiotic non-susceptibility phenotype. Other studies demonstrated high resistance rates of *V. alginolyticus* strains isolated from marine animals, with MDR strains showing the ability to transfer ARGs to susceptible strains [17,18]. Moreover, evidence that sponges support bacterial horizontal gene transfer was recently described [19]. Therefore, MDR *Vibrio* spp. strains are being hosted by marine invertebrates and may act in the dissemination of ARGs.

Regarding resistance phenotypes, *Vibrio* spp. are intrinsic carriers of the CARB family of penicillinases. This beta-lactamases cannot efficiently hydrolyze the early cephalosporins, such as cephalexin, and therefore produce a resistance phenotype [20]. Cephalexin-resistant strains were found to be negative, by means of PCR testing, for the tested beta-lactamase resistance genes, which could indicate that different (or new) mechanisms of cephalexin resistance could be involved in the observed phenotype. Furthermore, *Vibrio* can be intrinsically resistant to erythromycin since the outer membrane can act as a physical barrier that would make it difficult for the macrolide to enter the bacterial cell [21]. However, acquired macrolide resistance genes, which are important in medical practice, have been described in *Vibrio* spp. and are known to be carried by erythromycin-susceptible strains [4].

Interestingly, antibiotic non-susceptible strains occurred exclusively in Brazilian marine sponges, while plasmids occurred more frequently among strains isolated from the French sea urchins. The antibiotic resistance phenotype and plasmid occurrence correlated inversely. However, we cannot discard the possibility that the place of isolation of the marine organisms, a coastal region with high anthropogenic pressure, could influence the presence of these mobile genetic elements in *Vibrio* spp. Moreover, none of the isolates were positive for integron class 1, (*intI1*). Thus, our results suggest that the antibiotic resistance phenotypes were not related to plasmid and/or class 1 integron occurrence, as observed in another study [22]. This lack of relationship could indicate that resistance genotypes can be integrated into the chromosomes and/or into another type of integrative element. In *Vibrio* spp., superintegrons (~120 kb) are common and are dynamic in the acquisition and release of gene cassettes due to their IntIA integrase performance, which has 45 to 50% similarity with IntI1 to IntI3 [1]. Furthermore, *Vibrio* spp. plasmids may be related to other functions besides carrying ARGs, such as encoding antigenic variation to evade the host’s immune system, proteins or siderophores associated with iron uptake, and adaptation to environments contaminated by heavy metals [1,18].

Some of the highlights of the present study were the detection of *bla*_SHV_, *vhh* and *cusB* in the marine invertebrate-*Vibrio* spp. strains *V. alginolyticus* 84BHI10, *Vibrio* sp. ME7 and *V. oceanisediminis* P1I-2, respectively. The *bla*_SHV_ gene encodes a family of beta-lactamases with more than 220 members, according to the Beta-lactamase Database (http://www.bldb.eu/ accessed on 7 November 2021). These beta-lactamases are able to hydrolyze penicillins, extended-spectrum cephalosporins and monobactams, but are susceptible to inhibition by beta-lactamase inhibitors. However, the *bla*_SHV_ carrying strain demonstrated an intermediate resistance phenotype to ampicillin-sulbactam. A variant enzyme, SHV-10, that is not inhibited by beta-lactamase inhibitors has been described, but lost the ability to hydrolyze cephalosporins, showing catalytic activity only for penicillins [23]. Thus, the marine sponge-associated strain could be carrying the SHV-10 or a new SHV variant not inhibited by beta-lactamase inhibitors. Similar characteristics were described by another study, conducted by our group, on a *Shewanella* sp. strain isolated from the sea urchin *P. lividus* [24]. These reports configure the first descriptions of *bla*_SHV_ in bacteria from marine sponges and sea urchins, reinforcing the role of *Vibrio* spp. as ARGs’ reservoir in marine invertebrates.

The detection of *vhh* in *Vibrio* sp. ME7 from marine sponge is very interesting, because the *vhh* gene encodes a hemolysin protein, a potent virulence factor involved in pathogenicity from most *V. harveyi* in a range of fish and shellfish species [25]. The presence of this gene confirms the pathogenic potential of this isolate, whether *V. coralliilyticus* or *V. harveyi*, since both identification methods used presented low scores for species identification. However, considering *vhh* as a molecular marker of *V. harveyi* [25], there is a strong possibility of this strain belonging to this species. Alternatively, these low scores (<2.300) and identities (<95%) may indicate a novel *Vibrio* species. With regard to the closest species, *V. coralliilyticus* is a well-known pathogen of corals that is responsible for tissue lysis, bleaching and imperative losses of coral reefs worldwide [26]. Moreover, this species has shown the capacity to infect fish and oysters, with an enormous virulence gene repertoire, including encoders of hemolysins/cytolysins [27,28]. In the same line, *V. harveyi* represents a serious pathogen of marine animals, specially of cultured marine fish, causing diseases responsible for severe economic losses in the aquaculture industry [29]. Various virulence factors were already detected in *V. harveyi* strains isolated from lesions of marine fish, with high frequency of hemolysins/cytolysins (*vhh*, *vvh* and *hlyA*) [7,8]. Despite scarce information of this species leading to human infection, their pathogenic potential in marine animals is imperative, suggesting that *Vibrio* sp. ME7 has pathogenic capacity to cause human and sponge disease. However, more research is needed since ME7 was found to be erythromycin resistant. Therefore, this finding suggests that marine sponges can act as host reservoirs for antibiotic-resistant animal pathogens.

In relation to HMRGs, the *cusB* gene (copper and silver resistance encoder) was detected in the plasmid carrier *V. oceanisediminis* P1I-2 isolated from one sea urchin specimen (*P. lividus*). This specimen was the same as that from which *Shewanella* sp. cited above (also a *cusB* carrier) was isolated [10,24]. Thus, our results can suggest that the habitat of these sea urchins (Aber, France) may be contaminated by toxic metals, such as copper and/or silver, and that bacteria carrying heavy-metal resistance genes could be selected in these marine invertebrates.

The fact that other ARGs for a given phenotype were not detected by PCR in this work suggests that further antibiotic resistance mechanisms (which may be unknown) could be involved in the observed phenotypes. Similar observations of the non-detection of ARGs for a given phenotype have also been reported by other studies [24,30]. Such mechanisms, added to the resistance and virulence genes detected here, may represent (new) risks for public health, since *Vibrio* spp. of marine sponges and sea urchins could eventually act as donors of these genes to other bacteria from microbiota of these invertebrates. The close contact between them could allow this. Therefore, such genes could reach other bacteria and new habitats, such as humans and animals, leading to the worsening of the antibiotic resistance crisis [19].

## 4. Materials and Methods

### 4.1. Bacterial Isolates

Twenty-five *Vibrio* spp. were selected based on their viability and analyzed from a culture collection isolated from two phyla of marine invertebrate organisms. Nineteen strains were isolated from the Brazilian marine sponges *Darwinella* spp., *Haliclona* sp. and *Plakina cyanorosea* as described [13]. The other six strains were isolated from the gastrointestinal tract of the French sea urchin *Paracentrotus lividus* as described [10]. Briefly, all sponge-*Vibrio* spp. strains were isolated on brain heart infusion (BHI) agar and marine agar, tenfold diluted or not, while urchin-*Vibrio* spp. strains were isolated on marine agar or tenfold-diluted marine agar, and both mediums were supplemented with 10 μg/mL cycloheximide and incubated at 25°C [10,13]. The reactivation of strains for the present analysis occurred under the same conditions and using the same culture mediums as those of the original isolation. All isolates were previously identified by 16S rRNA sequencing as *Vibrio alginolyticus* (14), *Vibrio azureus* (1), *Vibrio gigantis* (1), *Vibrio harveyi* (1), *Vibrio ichthyoenteri* (1), *Vibrio oceanisediminis* (3) and *Vibrio* spp. (4) (Appendix A) [10,13].

### 4.2. MALDI-TOF MS Identification

*Vibrio* spp. isolates were also identified by Matrix-Assisted Laser Desorption/Ionization-time-of-flight mass spectrometry (MALDI-TOF MS) on a Microflex LT MS platform (Bruker Daltonics) with the samples being prepared as previously described [13]. The mass spectra obtained were compared to the references in the database using MALDI Biotyper 7.0 (Bruker^®^) software. Score values considered for identification were those recommended by the manufacturer: ≥2.300 indicated confidence at the species level; 2.299–2.000 at the genus level and likely for the species level; 1.700–1.999 solely at the genus level; and <1.699 was considered not reliable for identification.

### 4.3. Antibiotic Susceptibility Profiling

Antibiotic susceptibility was determined by means of the disc-diffusion method according to the Clinical and Laboratory Standards Institute [31]. Twenty-three antibiotic discs (Sensidisc, São Paulo, Brazil) of the β-lactams (amoxicillin-clavulanic acid (30 µg), ampicillin-sulbactam (20 µg), piperacillin-tazobactam (110 µg), aztreonam (30 µg), cefepime (30 µg), cefoxitin (30 µg), ceftazidime (30 µg), ceftriaxone (30 µg), cephalexin (30 µg), ertapenem (10 µg), imipenem (10 µg), meropenem (10 µg)), aminoglycosides (gentamicin (10 µg), amikacin (30 µg), tobramycin (10 µg)), fluoroquinolones (nalidixic acid (30 µg), ciprofloxacin (5 µg), levofloxacin (5 µg), ofloxacin (5 µg)), folate metabolism inhibitor (trimethoprim-sulfamethoxazole (1.25–23.75 μg)), macrolide (erythromycin (15 µg)), phenicol (chloramphenicol (30 µg)), and tetracycline (tetracycline (30 µg)) classes or groups were selected. The quality control strains *Escherichia coli* ATCC^®^ 25922 and *Escherichia coli* ATCC^®^ 35218 (for β-lactam/β-lactamase inhibitor discs), and the susceptibility interpretation criteria used were as described in CLSI [31] with modifications according to da Costa et al. [24]. Multidrug-resistance (MDR) was defined as acquired non-susceptibility to at least one agent of three or more antibiotic categories [32]. The heteroresistance phenotype was characterized when a purified strain showed growth of resistant colonies within the inhibition zone [33].

### 4.4. Genomic and Plasmid DNA Extractions

Genomic DNA was obtained using the UltraClean Microbial DNA Isolation kit (Mo Bio, Carlsbad, CA, USA) and stored at −20 °C until use for PCR amplification. Plasmid DNA was prepared employing Wizard Plus SV Minipreps DNA Purification Systems (Promega, Madison, WI, USA). *Klebsiella pneumoniae* Kp13 plasmids [34] were used as molecular size markers for plasmids.

### 4.5. PCR Assays

Strains were screened by PCR amplification for ARGs encoding resistance to beta-lactams (*bla*_TEM_, *bla*_SHV_, *bla*_GES_), aminoglycosides (*aac(6′)-Ie-aph(2′)-Ia*, *aac(6′)-Ib, ant(2”)-I*)*,* macrolides (*mef*(C), *mph*(G), *ermB, ereA, mphA, mefA*) and quinolones (*qnrA*, *qnrB*, *qnrC*, *qnrD*, *qnrS*, *qnrVC*) for stains that showed the respective resistance phenotypes. At the same time, all 25 strains were screened by PCR amplification for HMRGs encoding resistance to cadmium (*cadA*), copper (*czcB*, *cusB*), lead (*pbrA*) and mercury (*merA*), virulence genes encoding hemolysins (*toxR*, *tdh, trh*, *vhh*) and type VI secretion systems (*T6SS*), and genes encoding the integrase from Class 1 integron (*intI1*). Multiplex PCR reactions were performed for ARGs encoding resistance to beta-lactams (beta-lactamase-coding genes) and quinolones (Qnr-coding genes) [24].

PCR was conducted in a total volume of 25 μL as described [24]. All primer sequences and amplification conditions were taken from their respective references (Appendix A). Genomic DNA from strains of *Vibrio parahaemolyticus* IOC 17381 (*toxR*^+^ and *tdh*^+^) and 0798081 (*trh*^+^) were used as positive controls for the respective virulence genes targeted by PCR [35]. Amplicons were observed through 1–2% agarose gel electrophoresis using the Low DNA Mass Ladder (Thermo Fisher Scientific, Waltham, MA, USA) as the molecular size standard.

### 4.6. Verification of Antibiotic Resistance Phenotype and Plasmids Association

The correlation between the antimicrobial resistance phenotype and the presence of plasmids was assessed for all strains together and for the 15 potentially pathogenic strains (*V. alginolyticus* and *V. harveyi*) separately. The Pearson’s correlation coefficients were calculated with 95% confidence intervals using GraphPad Prism software, version 8.0.2.

## 5. Conclusions

With the advance of the antimicrobial resistance crisis, non-cholera *Vibrio* strains are being investigated as resistance gene reservoirs in aquatic environments. Our study revealed that marine sponges and sea urchins in natural environments are harboring antimicrobial-resistant *Vibrio* spp., which carry important and maybe new antimicrobial and heavy metal resistance genes, as well as virulence genes and plasmids. These findings corroborate that *Vibrio* spp. associated with marine sponges and sea urchins may pose a potential risk to public health. These aquatic animals can act as an antimicrobial-resistant- and pathogenic-*Vibrio* spp. reservoir in the environment. Further investigations may reveal antimicrobial resistance genotypes that are not possible to detect using PCR methodologies and demonstrate the genetic environment of ARGs and virulence genes, thereby increasing the analytic power of the resistome, virulome and mobilome of non-cholera *Vibrio* strains from the environment and leading to potential further consequences for public health.

## Figures and Tables

**Figure 1 antibiotics-10-01561-f001:**
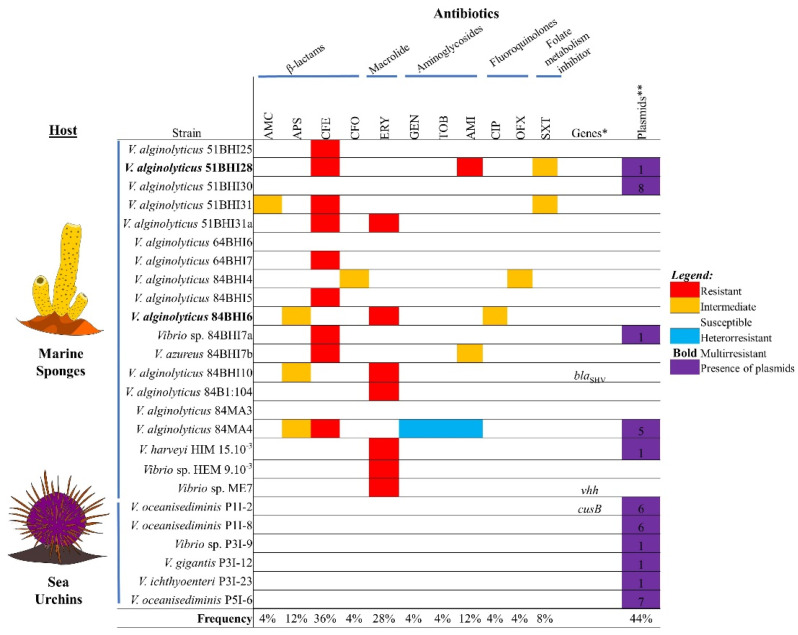
Non-susceptibility frequency of *Vibrio* spp. strains from marine sponges and sea urchins. The isolates were classified as resistant, multirresistant, intermediate, susceptible or heteroresistant to antibiotics. The antibiotic and heavy metal resistance genes, virulence genes (*) and number of plasmid forms (**) detected were shown. AMC: amoxicillin-clavulanic acid; AMI: amikacin; APS: ampicillin-sulbactam; CFE: cephalexin; CFO: cefoxitin; CIP: ciprofloxacin; ERY: erythromycin; GEN: gentamicin; OFX: ofloxacin; SXT: trimethoprim/sulfamethoxazole; TOB: tobramycin.

## Data Availability

Not applicable.

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
