# Peer review of "Antibiotic and Heavy Metal Susceptibility of Non-Cholera Vibrio Isolated from Marine Sponges and Sea Urchins: Could They Pose a Potential Risk to Public Health?"

_antibiotics, 2021, doi:10.3390/antibiotics10121561_

Round 1

Reviewer 1 Report

The paper is well built and merits publication in its present form, minor modifications could be proposed, but that only would delay publication. Vibrio microbiologically speaking is a complex genus, and the results supporting the conclusion that any species of a given bacterial genus  could be the reservoir for antibitiotic resistant genes should be published, according to my criteria

Author Response

Reviewer 1

The paper is well built and merits publication in its present form, minor modifications could be proposed, but that only would delay publication. Vibrio microbiologically speaking is a complex genus, and the results supporting the conclusion that any species of a given bacterial genus could be the reservoir for antibitiotic resistant genes should be published, according to my criteria

Answer: Thank you very much for the considerations.

Reviewer 2 Report

Overview and general recommendations:

Overall, the study is well designed, the experiments are well performed and the results are clearly presented.  

However, some indications for minor revision are given below.

Line 87: "... researched in this study" may be replaced by "investigated in this study". It is more adequate.

Line 135: "Here" maybe replace by "Herein", it is more suitable.

Line 135: "...as potential pathogen, it was...". You should eliminate "it" to give a sens to the sentence.

Line 191: " ... this specie has been shown the capacity...". You should delete the passive form.

Line 193: "In the same line, V. harveyi is a serious pathogen..".Try to change "is" by "represents". It is more adequate.

The introduction may be improved.

Is there any information about Sampling of the Brazilian marine sponges and French sea urchin?

Try to develop the conclusion. Are there any fallouts and/or perspectives of the investigation?

Spell check of english language is required.

Author Response

Reviewer 2

Overview and general recommendations:

Overall, the study is well designed, the experiments are well performed and the results are clearly presented.  

However, some indications for minor revision are given below.

Line 87: "... researched in this study" may be replaced by "investigated in this study". It is more adequate.

Line 135: "Here" maybe replace by "Herein", it is more suitable.

Line 135: "...as potential pathogen, it was...". You should eliminate "it" to give a sens to the sentence.

Line 191: " ... this specie has been shown the capacity...". You should delete the passive form.

Line 193: "In the same line, V. harveyi is a serious pathogen..".Try to change "is" by "represents". It is more adequate.

The introduction may be improved.

Answer: The entire manuscript has been carefully revised and improved.

Is there any information about Sampling of the Brazilian marine sponges and French sea urchin?

Answer: Due to the limited number of words and space in the text and as this information was cited by the group's previous articles, in this manuscript we present it as references:

  1. Laport, M.S.;Bauwens, M.; Collard, M.; George, I. Phylogeny and antagonistic activities of culturable bacteria associated with the gut microbiota of the sea urchin (Paracentrotus lividus). Curr. Microbiol. 2018, 75, 359-367, doi:10.1007/s00284-017-1389-5
  2. Freitas-Silva, J.; Silva-Oliveira, T.; Muricy, G.; Laport, M.S. Bacillus strains associated to Homoscleromorpha sponges are highly active against multidrug resistant bacteria. Curr. Microbiol. 2020, 77, 807-815, doi:10.1007/s00284-019-01870-x

Try to develop the conclusion. Are there any fallouts and/or perspectives of the investigation?

Answer: Thank you, the conclusion was developed and perspectives were included.

Spell check of english language is required.

Answer: The entire manuscript has been carefully revised and improved.

Reviewer 3 Report

  1. In line 12, 'composing' microbiota does not seem to be the right word
  2. In material methods section "Bacterial isolates", mention media and culture condition briefly.
  3. In material methods section "Antibiotic susceptibility profiling", mention antibiotics used with their class/group.
  4. In PCR assay, mention genes used for PCR.
  5. Line 120 is not complete
  6. PCR targeting some species specific gene should be performed, for example for the isolate which contained vvh. If performed, PCR results targeting at least the big four Vibrios should be mentioned perhaps in a table.

Author Response

Reviewer 3

  1. In line 12, 'composing' microbiota does not seem to be the right word

Answer: The entire manuscript has been carefully revised and improved and this line was rewritten: “… part of the microbiota from marine invertebrates”.

2. In material methods section "Bacterial isolates", mention media and culture condition briefly.

Answer: All considerations were incorporated in the manuscript.

          “Briefly, all sponge-Vibrio spp. strains were isolated on brain heart infusion (BHI) agar and marine agar, tenfold diluted or not, while urchin-Vibrio spp. strains were isolated on marine agar or tenfold diluted marine agar, both mediums supplemented with 10 μg/ml cycloheximide and incubation at 25°C [10,13]. The reactivation of strains for the present analysis occurred on the same conditions and culture mediums from the original isolation.”.

3. In material methods section "Antibiotic susceptibility profiling", mention antibiotics used with their class/group.

Answer: All considerations were incorporated on the manuscript.

            “… β-lactams [amoxicillin-clavulanic acid (30 µg), ampicillin-sulbactam (20 µg), piperacillin-tazobactam (110 µg), aztreonam (30 µg), cefepime (30 µg), cefoxitin (30 µg), ceftazidime (30 µg), ceftriaxone (30 µg), cephalexin (30 µg) , ertapenem (10 µg), imipenem (10 µg), meropenem (10 µg)], aminoglycosides [gentamicin (10 µg), amikacin (30 µg), tobramycin (10 µg)], fluoroquinolones [nalidixic acid (30 µg), ciprofloxacin (5 µg), levofloxacin (5 µg), ofloxacin (5 µg)], folate metabolism inhibitor [trimethoprim-sulfamethoxazole (1.25-23.75 μg)], macrolide [erythromycin (15 µg)], phenicol [chloramphenicol (30 µg)], and tetracycline [tetracycline (30 µg)] classes or groups were selected.”.

4. In PCR assay, mention genes used for PCR.

Answer: All considerations were incorporated on the manuscript.

            “… beta-lactams (blaTEM, blaSHV, blaGES), aminoglycosides (aac(6′)-Ie-aph(2′)-Ia, aac(6′)-Ib, ant(2”)-I), macrolides (mef(C), mph(G), ermB, ereA, mphA, mefA) and quinolones (qnrA, qnrB, qnrC, qnrD, qnrS, qnrVC) for stains that shown the respective resistance phenotype. At the same time, all 25 strains were screened by PCR amplification for HMRGs encoding resistance to cadmium (cadA), copper (czcB, cusB), lead (pbrA) and mercury (merA), virulence genes encoding hemolysins (toxR, tdh, trh, vhh) and type VI secretion system (T6SS), and gene encoding the integrase from Class 1 integron (intI1)”.

5. Line 120 is not complete

Answer: The consideration was incorporated in the manuscript.

            “… https://gtdb.ecogenomic.org/stats/r95. Accessed on 09 December 2021”.

6. PCR targeting some species specific gene should be performed, for example for the isolate which contained vvh. If performed, PCR results targeting at least the big four Vibrios should be mentioned perhaps in a table.

Answer: Dear reviewer, some virulence genes tested are species-specific. For example, the vhh gene for hemolysin is species-specific for pathogenic Vibrio harveyi; toxR, tdh, and trh are Vibrio parahaemolyticus species-specific. Especially, toxR is a molecular marker for the O3:K6 pandemic clone of that species. As the genus Vibrio is complex to taxonomic identification, mainly for at the species level. We obtained low identity (16S rRNA) and score (MALDI-TOF) for some strains, and we chose to test the 25 Vibrio spp. Strains for the five virulence genes, the four heavy metal resistance genes and the intI1 gene for integrase from class 1 integron. Thus, we were looking at the possibility of amplification of these genes in Vibrio genus strains isolated from marine sponges and sea urchins. Consistently, we found only the vhh gene in the strain ME7 that showed low identification (94.45%) by 16S rRNA sequencing and score with reliability for genus and likely for Vibrio harveyi species (2,179) by MALDI-TOF analysis. Therefore, considering that vhh gene is used as a molecular marker for pathogenic V. harveyi by some studies (Conejero and Hedreyda, 2004; Sun et al., 2009), and the results from MALDI-TOF, we believe that ME7 belongs to the V. harveyi species. However, as identification by 16S rRNA sequencing shows low identity (< 98.7%), we still remain reluctant to classify ME7 as a V. harveyi. The other strains were PCR negative for the investigated virulence genes, suggesting that: (i) either the strains do not belong to the species V. harveyi or V. parahaemolyticus; (ii) or they do not carry these genes, (iii) or even, some strains can carry the investigated genes, but show alterations in the gene sequence and thus hindering primer hybridization. Furthermore, (iv) it could be that the strains which are PCR-negative for virulence genes do not belong to pandemic clones, such as O3:K6 of V. parahaemolyticus. However, the possibility that part of these bacteria are pathogenic is not ruled out. But the information about which strains were tested for specific genes were incorporated in the section Materials and Methods.

References cited on the answer:

  • Conejero, M.J.U.; Hedreyda, C.T. PCR detection of hemolysin (vhh) gene in Vibrio harveyi. Gen. Appl. Microbiol. 2004, 50, 137-142, doi:10.2323/jgam.50.137
  • Sun K, Hu YH, Zhang XH, Bai FF, Sun L. Identification of vhhP2, a novel genetic marker of Vibrio harveyi, and its application in the quick detection of harveyi from animal specimens and environmental samples. J. Appl. Microbiol. 2009, 107, 1251-1257, doi:10.1111/j.1365-2672.2009.04304.x